# Is ALDH1L1 Elevated in Lung Cancer? Comment on: Lee, S.-H.; et al. “The Combination of Loss of ALDH1L1 Function and Phenformin Treatment Decreases Tumor Growth in KRAS-Driven Lung Cancer” *Cancers* 2020, *12*, 1382

**DOI:** 10.3390/cancers13071691

**Published:** 2021-04-02

**Authors:** Sergey A. Krupenko, Jaspreet Sharma

**Affiliations:** Department of Nutrition and Nutrition Research Institute, University of North Carolina at Chapel Hill, Kannapolis, NC 28081, USA; sharmaj@email.unc.edu

We read with interest the article by Lee et al. [1], which evaluated the folate enzyme ALDH1L1 as a therapeutic target in the treatment of non-small cell lung cancer. The authors hypothesized that upregulation of ALDH1L1 expression is linked to KRAS, and they performed a set of experiments to test this hypothesis in cell culture and animal models. In the initial experiment, the authors demonstrated the expression of ALDH1L1 protein in A549 lung cancer cells, a highly surprising finding. Indeed, our research group has shown in 2002 [2] that A549 cells (as in the majority of cancer cell lines) do not express this protein. In agreement with our data, the Human Protein Atlas (https://www.proteinatlas.org/ENSG00000144908-ALDH1L1/cell; accessed on 15 August 2020) also shows a lack of *ALDH1L1* mRNA in A549 cells. We later established that silencing of the *ALDH1L1* gene in several cancers and cancer cell lines (including A549 cells) is associated with strong methylation of the gene promoter [3].

We became curious as to why such a disagreement with earlier published data appeared in Lee’s paper. To make sure that the appearance of the band assigned as ALDH1L1 protein in Lee’s paper is not associated with abnormal properties of cells used in the study, or by specificity of a particular antibody, we obtained A549 cells from the same source, the DCTD Tumor Repository, NCI/NIH, and tested them for the presence of ALDH1L1 using an Abcam antibody (cat. #ab175198), as in Lee’s paper. As can be seen in the figure, there was no detectable ALDH1L1 protein in the A549 cells. As a control, we used RT4 cells known to express high levels of this protein. We also used our in-house ALDH1L1-specific antibody, which has been validated and used in numerous of our published studies, as well as in studies from other research groups [2,4,5,6]. This antibody did not detect any ALDH1L1 protein in the A549 cells either (Figure 1). Further, in agreement with our previous report [7], levels of *ALDH1L1* mRNA evaluated by qPCR were below the detection limit in the A549 cells (Figure 1). We also re-evaluated H1299 cells for the ALDH1L1 expression, since we previously reported a lack of the protein in these cells [8]. Neither ALDH1L1 protein nor its mRNA were detected in the H1299 cells (Figure 1).

Surprisingly, Lee and the co-authors failed to acknowledge that there is a significant body of literature demonstrating a strong downregulation of ALDH1L1 in several cancers (recently reviewed in [9]). The only paper suggesting an upregulation of the protein in a specific cancer, lung cancer, was from the same group [10]. Curiously, the paper reported that A549 cells have the highest levels of ALDH1L1 protein among several cell lines [10]. It should be also noted that Lee’s paper [1] contains several incorrect statements regarding the published literature. In the introduction to their study, the authors cited two papers [11,12] to suggest that ALDH1L1 is upregulated in lung cancer, but neither of the references show an upregulation of ALDH1L1 in non-small cell lung cancer. In fact, these papers did not even mention ALDH1L1. Further, the domains of ALDH1L1 are not listed correctly. Curiously, throughout the paper, the authors emphasized the production of NADH by ALDH1L1. ALDH1L1, however, has exceptionally strong preference for NADP^+^ [13], and there is no experimental evidence that the enzyme can utilize NAD^+^ in vivo. Lee and the co-authors made their speculation based on the ratios of NAD^+^/NADH versus NADP^+^/NADPH, which is not a sufficient argument (we refer readers to an excellent review on the subject [14]).

Considering that the information presented by the authors on the expression of ALDH1L1 in A549 and H1299 cell lines contradicts the previously published expression pattern of the protein, as well as their failure to acknowledge commonly observed ALDH1L1 downregulation in cancers and their unsubstantiated statements with incorrect citations, we suggest that this paper needs to be revised in order to prevent misleading the scientific readership.

## Figures and Tables

**Figure 1 cancers-13-01691-f001:**
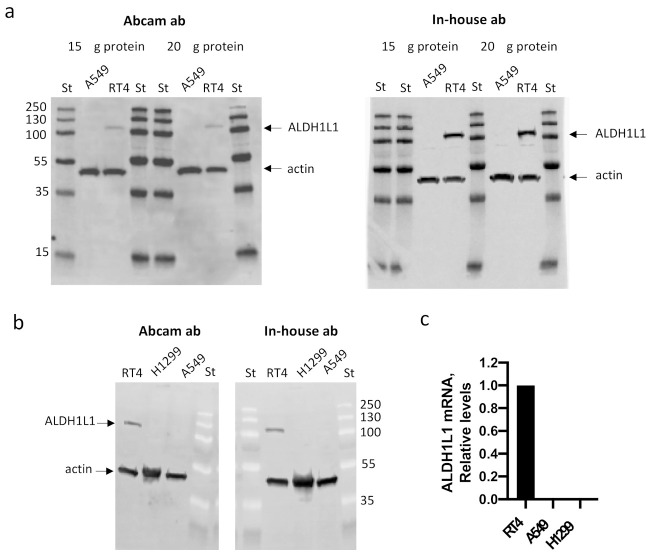
Levels of ALDH1L1 protein (**a**,**b**, Western blot) and mRNA (**c**, qPCR) in A549, H1299, and RT4 cells. The Abcam antibody cat. #ab175198 (used in Lee’s paper [1]), as well as our in-house antibody [2], did not detect ALDH1L1 in the A549 or H1299 cells. Actin was used as loading control. Twenty µg of total protein was loaded in panel b. St, pre-stained protein ladder (ThermoFisher, cat. #26619; numbers indicate molecular masses of standards).

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
