# Peer review of "Is ALDH1L1 Elevated in Lung Cancer? Comment on: Lee, S.-H.; et al. “The Combination of Loss of ALDH1L1 Function and Phenformin Treatment Decreases Tumor Growth in KRAS-Driven Lung Cancer” Cancers 2020, 12, 1382"

_cancers, 2021, doi:10.3390/cancers13071691_

Round 1
Reviewer 1 Report
- It’s recommended to provide detailed information (e.g. sequences of PCR primers and validation data of the in-house antibody) of experimental materials used in the present comment.
- In Fig 1a and 1b. The patterns of pre-stained protein ladders are so different.
- Fig 1c is a quantitative result of ALDH1L1 mRNA. However, the figure legend was described as “twenty μg of total protein was loaded in panel c”.
Reviewer 2 Report
The comment made on the article Lee, S.-H.; et al. “The Combination of
Loss of ALDH1L1 Function and Phenformin
Treatment Decreases Tumor Growth in KRAS-Driven
Lung Cancer” Cancers 2020, 12, 1382. is interesting.
- I agree with the authors as the original article did not perform antibody specificity experiments.
- The molecular size of the protein is missing in the western blots that was presented in the original article.
- Most of the data is based on the IHC data in the article.
Based on the data shown in Sergey Krupenko and Jaspreet Sharma comment article. I would suggest the following to strengthen the comment.
- The authors could choose ALDH1L1 expressing RT-4 cell line, and perform Knockout experiments with an siRNA show the knockout and specificity of the protein.
- As a rescue experiment, the authors can reintroduce ALDH1L1 in the knockout cells or another cell line that is deficient of ALDH1L1.
- It would interesting to know the ALDH1L1 mRNA expression in these cells.
By this way the authors could show the specificity of the ALDH1L1 antibody.
